# MAVEN2: An Updated Open-Source Mass Spectrometry Exploration Platform

**DOI:** 10.3390/metabo12080684

**Published:** 2022-07-25

**Authors:** Phillip Seitzer, Bryson Bennett, Eugene Melamud

**Affiliations:** Calico Life Sciences, LLC, South San Francisco, CA 94080, USA; bryson@calicolabs.com (B.B.); eugene@calicolabs.com (E.M.)

**Keywords:** metabolomics, lipidomics, software, open-source, fragmentation, visualization, identification, GUI

## Abstract

MAVEN, an open-source software program for analysis of LC-MS metabolomics data, was originally released in 2010. As mass spectrometry has advanced in the intervening years, MAVEN has been periodically updated to reflect this advancement. This manuscript describes a major update to the program, MAVEN2, which supports LC-MS/MS analysis of metabolomics and lipidomics samples. We have developed algorithms to support MS/MS spectral matching and efficient search of large-scale fragmentation libraries. We explore the ability of our approach to separate authentic from spurious metabolite identifications using a set of standards spiked into water and yeast backgrounds. To support our improved lipid identification workflow, we introduce a novel in-silico lipidomics library covering major lipid classes and compare searches using our novel library to searches with existing in-silico lipidomics libraries. MAVEN2 source code and cross-platform application installers are freely available for download from GitHub under a GNU permissive license [ver 3], as are the in silico lipidomics libraries and corresponding code repository.

## 1. Introduction

Rapid advances in the field of mass spectrometry call for a continued evolution of computational methods to adapt to improved instrumentation and novel workflows. In the past decade, instruments capable of high mass resolution combined with parallel reaction monitoring (PRM) modes have become widely available [1,2]. MAVEN was originally developed to process full scan metabolomics data to support isotopic tracing and general profiling workflows [3]. Over time, the program was adopted by academic and industrial labs and has evolved to support the specific needs of these organizations. The codebase has expanded, and along with its cousin EL-MAVEN [4], provides the community with algorithms to deal with complex data types and larger mass spectrometry datasets.

Adapting to emerging data modalities, increased throughput, and other recent improvements in the field of mass spectrometry, new data processing algorithms and data interrogation tools have been developed in MAVEN. Specifically, the advent of tandem mass spectrometry (MS/MS) led us to develop a novel MS/MS-based peak group formation approach and implement compound identification by MS/MS spectral library matching. Identification of compounds by spectral library search has been shown to be an effective identification approach for metabolomics, lipidomics, and proteomics datasets [5,6,7,8]. Simultaneously, we have implemented the ability to easily examine the data by manually investigating peak grouping, MS and MS/MS spectra, and potential matches between observed data and library compounds.

In this report, we present MAVEN2: an expanded tool for effective analysis and interrogation of MS/MS metabolomics and lipidomics data. The software has a carefully designed user interface for analyzing and reviewing large datasets collected with MS/MS spectra. In addition to releasing the MAVEN2 application, we are also releasing open-source code for generation of in silico lipidomics libraries. Our libraries are built by collecting fragmentation spectra from commercially available internal standards and theoretically extending these spectra to similar compounds with different acyl chain lengths, as well as using published reference spectra [9]. In our evaluation, we found a number of lipid classes where the CalicoLipids in silico library returned more identifications than other prominent open-source in silico lipidomics libraries, suggesting that our lipid spectral library may be a useful resource for researchers interested in enumerating the lipidome.

## 2. Materials and Methods

### 2.1. Informatic Methods

#### 2.1.1. MS/MS-Based Peak Grouping and Compound Identification

In MAVEN2, we have added an MS/MS-based peak grouping and compound identification approach (Figure 1A). Our approach requires that at least one peak in a peak group be associated with an MS/MS scan. This MS/MS association requirement yields higher quality identifications [10] and reduces computation time by filtering the MS1 (*m*/*z*, retention time) search space prior to peak detection. Initially, MS/MS scans are processed to form “slices”, or blocks in (*m*/*z*, retention time) space, centered around MS/MS precursor *m*/*z* and scan retention time values (Figure 1B). All slices are progressively merged if they overlap in both *m*/*z* and retention time range, regardless of the sample from which they derive, until no slices overlap (Figure 1C). Following this merging step, peak grouping is carried out as previously described [3] (Figure 1D). The MS/MS scans associated with the pre-merged slices are assigned to the finalized peak groups based on retention time proximity. Once peak groups have been formed, spectral matching can be performed to annotate peak groups as specific compounds (Figure 1E). In cases where peak groups fail to match to any library entries, they are either discarded or retained as unannotated peak groups (depending on the user’s preference).

The implementation of this workflow necessitated the development of additional infrastructure to manage various spectral libraries. Spectral libraries may be imported into MAVEN2 in csv or NIST msp format and stored in an on-disk SQLite database. Popular freely available libraries for metabolomics and lipidomics, such as MoNA, LipidBlast, and MS-DIAL, are available in NIST msp format and may therefore be loaded into MAVEN2. Spectral libraries may be very large with millions of spectra. We have optimized the MAVEN2 codebase to support caching and efficient searching of large libraries. We have also implemented the ability to browse the contents of imported spectral libraries, including the library spectra themselves. 

#### 2.1.2. MS/MS-Based Spectral Matching and Scoring Approaches

Compared to MS1-only matching, MS/MS-based matching significantly improves identification confidence [10]. Spectral library searching has been used extensively for analysis of LC-MS/MS metabolomics and lipidomics datasets for a number of years, and has been implemented in some form in a number of open-source and closed-source analysis tools, such as XCMS [11], Elements for Metabolomics [12], MS-DIAL [13], LipidSearch [14], LipidExplorer [15], CompoundDiscoverer [16], and others [17,18].

Many scoring approaches have already been developed and described elsewhere [5,19]. In MAVEN2, we have implemented several well-established scoring algorithms such as the number or fraction of matched reference spectral fragments, the dot product score (also known as the cosine score), and the percentage of reference spectrum TIC matched. We also adopted matching algorithms from proteomics, such as hypergeometric score (HGS) and multiple class weighted hypergeometric score (MVH) [20]. The hypergeometric score we have implemented is similar to the approach used in the proteomics search tool MyriMatch [20], which uses hypergeometric distributions to model peptide-spectrum matches. Briefly, hypergeometric distributions evaluate spectral similarity by comparing the number of observed matched fragments to the likelihood that two fragments might match by chance. If an observed spectrum contains m fragments, a reference spectrum n fragments, k fragment matches between them, and a reference spectrum can be divided into N non-overlapping *m*/*z* buckets where a spectral match might occur, the hypergeometric score is given by (Equation (1)).
(1)$$\mathrm{Score}(k,\;m,\;n,\;N)=-\ln\left(\frac{\left(\begin{array}{c} m\\ k \end{array}\right)\times \left(\begin{array}{c} N-m\\ n-k \end{array}\right)}{\left(\begin{array}{c} N\\ n \end{array}\right)}\right)$$

N is tied to the resolving power associated with the MS/MS spectra and should be set such that two spectral peaks that ought to match together are placed in the same *m*/*z* bucket. When applied to high resolution mass spectrometry data (>50,000 RP), N should be a large number, on the order of 1 × 10^5^ or higher.

The hypergeometric score does not include any comparison of spectral intensity of matched fragments. This is useful in situations where spectral libraries lack expected fragment intensity information or where library spectra contain very different intensity values than the observed data (for example, a reference spectrum is collected using a different collision energy than the observed spectrum). Furthermore, if an observed MS/MS spectrum is chimeric, the hypergeometric score may still be used to identify a strong match, as the unmatched spectral peaks associated with a different species do not overly penalize the matching algorithm. Other scoring approaches that rely on intensity comparisons (such as the dot product score, for instance) do not perform well when applied to chimeric spectra.

#### 2.1.3. Lipidomics Search Compound Comparison Approach

All lipidomics samples were searched in MAVEN2 against both the CalicoLipids spectral library and the MS-DIAL spectral library, which was retrieved from [13]. The hypergeometric score was used to assess spectral quality. A complete list of all search parameters is available in (Appendix A). Though both spectral libraries attempt to list fragments associated with specific lipid species, we found that the MS-DIAL spectral library in general listed fewer fragments per compound than the CalicoLipids spectral library. When the libraries were searched against lipidomics samples, the difference in library construction between the CalicoLipids and MS-DIAL libraries resulted in systematic differences in the distribution of match scores associated with each library.

Given the difficulty of estimating a false discovery rate for identifications in metabolomics and lipidomics, we were interested in (1) determining a score threshold that could be used as a reasonable surrogate for identification accuracy, and (2) aligning search results across the two libraries. The lack of gold standards in our data, combined with systematic differences in species fragment count between the two libraries, required us to develop a novel approach to effectively compare annotations between the two libraries (Appendix A). Our approach was motivated by comparing annotation histograms of agreements (CalicoLipids and MS-DIAL libraries suggest the same annotation for a given feature) and disagreements (CalicoLipids and MS-DIAL suggest different annotations for a given feature), organized by the hypergeometric score associated with feature—library matches (Appendix A). We found that collapsing agreements and disagreements into a single metric produced unimodal distributions for both the CalicoLipids and MS-DIAL libraries (Appendix A), each with a single global maximum, which we term the Retention Balance Point (RBP) (Equation (2)).
(2)$$\mathrm{RBP}=\mathrm{MAX}\left(\frac{num\;agreements\;\geq \;threshold}{all\;agreements}-\frac{num\;disagreements\;\geq \;threshold}{all\;disagreements}\right)$$

In our comparisons of lipidomics samples, we determined the RBP values to be 13.9 for the MS-DIAL spectral library and 22.8 for the CalicoLipids spectral library (Appendix A). Using these library-specific score thresholds as our criterion for annotation acceptance or rejection, we discarded compound annotations from a spectral library when the associated hypergeometric score value fell below the library-specific RBP value. In our workflow, if a feature was annotated by both libraries but fell below the library-specific score threshold for one library, only the annotation from the other library was retained, so this feature would be classified as having been successfully annotated by only one library.

#### 2.1.4. Generation of In Silico Lipid Spectral Library

Lipidomics data, like metabolomics data, is often generated using a data-dependent acquisition workflow, making it highly amenable to analysis in MAVEN2. To extend the capabilities of MAVEN2 into the field of lipidomics, we developed a python package defining rules to create a library of predicted fragmentation patterns covering 61 lipid classes and 15 adduct forms, specifically designed to mimic HCD fragmentation spectra at specific collision energies. Scripts in our package can be executed to produce spectral libraries formatted as one or more msp files, which may be parsed by MAVEN2. Our python package is publicly available on github at https://github.com/calico/CalicoLipidLibrary (accessed 23 July 2022).

Lipid fragmentation libraries were generated based upon a mixture of literature reports of lipid fragmentation, and fragmentation of lipid standards injected via direct infusion on a Thermo Q-Exactive Plus mass spectrometer (Thermo Scientific, San Jose, CA, USA). When injecting standards, we produced fragmentation spectra at a variety of collision energies, then summed the peak intensities from the spectra generated via HCD fragmentation. We then chose to build an in silico library based upon a linear combination of the spectra generated at 20, 30, and 40 NCE, allowing us to mimic the fragmentation pattern from a stepped collision energy setting for 20, 30, 40 NCE (see Appendix A for standards used). For lipid classes without an available chemical standard, we used the literature reports of fragmentation to predict masses of peaks we expected to see, but did not predict fragment intensities. We manually examined the spectra and assigned spectral peaks to the reaction we believed created them. The average intensity for each peak assignment across all the standards for that class was then used to make the in silico library. We generated fragmentation patterns for multiple adducts per lipid, typically [M+H]+ and [M+Na]+ in positive mode, and [M−H]− and [M+FA−H]− (formic acid) in negative mode. Fragmentation patterns for other adducts were created assuming that the fragmentation of positive mode adducts ([M+K]+, [M+NH4]+) were similar to [M+Na]+, and other negative mode adducts ([M+Cl−], [M+AcOH−H]− (acetate)) were similar to [M+FA−H]− adducts. We included spectral peaks which we could annotate, down to an intensity of 0.2% of the most abundant peak in the spectrum, as we found that these low intensity peaks were of value in correctly annotating the acyl composition of some lipids.

We have deposited msp files of our spectral library into the Metabolights database [21], with accession number MTBLS3097.

### 2.2. Experimental Methods

#### 2.2.1. Construction of Polar Metabolites Spectral Library

Metabolite spectra were generated using a Thermo Vanquish LC connected to a Thermo Q-Exactive Plus mass spectrometer (Thermo Scientific, San Jose, CA, USA). Pure chemical standards were dissolved in appropriate solvent, and loop injections into the mass spectrometer were performed, where we acquired fragmentation spectra at a variety of normalized collision energies. Libraries were created by summing the spectra from the three relevant energies (20, 40, and 80 NCE settings in positive mode, and 20, 50, and 100 NCE in negative mode) to match the data acquisition, using stepped collision energies. We have made these libraries publicly available as msp files (Appendix A).

#### 2.2.2. Generation of Metabolomics Datasets 

Metabolomics datasets consisted of chemical standards spiked into either a water or yeast background (i.e., no matrix or yeast matrix). Chemical standards were purchased and added to background in different combinations to form pooled samples. We created 25 pools of chemical standards, each containing 37 distinct metabolites. These 25 pools were combined together in different ways to form “super pools”. In total, 10 super pools were generated. Each pool was added to at least two different super pools. Each super pool sample contained 185 distinct metabolites. These 10 super pool samples were analyzed by LC-MS without matrix (water background), then analyzed again in yeast extract matrix (yeast background).

Yeast extract was generated by growing a single colony of S. cerevisiae DBY12000 in 10 mL of YPD medium overnight to saturation, then diluting the culture in 100 mL of YPD, grown to an OD650 of 0.9. Ten replicates of 2.75 mL of culture were collected by vacuum filtration, then immediately placed in an 80:20 methanol:water solution maintained at −20 °C, followed by centrifugation. The aspirants were combined into two large aliquots, then dried under nitrogen, and resuspended in either water for reverse phase chromatography or 80:20 acetonitrile:water for analysis via HILIC chromatography.

#### 2.2.3. Generation of Lipidomics Datasets

Male C57Bl/6J mice at 12 weeks of age were fasted for 4 h. The mice were anesthetized and retro-orbital bleeds of approximately 200 µL blood volume were performed, and samples were transferred into K2-EDTA tubes, mixed briefly and stored on ice. Samples were then centrifuged for 10 min at 4500× *g*. Plasma was aliquoted to 50 µL per tube and frozen on dry ice, and stored at −80 °C until extraction.

Lipids were separated via methyl-tertbutyl ether (MTBE) extraction, modified from (Matyash 2008). Plasma aliquots were transferred to 8 mL glass culture tubes with a PTFE lined cap. Extraction was performed by adding 1 mL of 50:50 methanol:water, followed by addition of 1 mL of MTBE. Samples were mixed using a vortex mixer for 1 min, then centrifuged at 3000× *g* for 5 min. The upper, MTBE-containing layer, was moved to a separate glass tube, followed by a second extraction with another 1 mL of MTBE. The second MTBE extraction was combined with the first, and then dried under nitrogen. Samples were resuspended in 200 uL of 1:1:2 water:methanol:butanol containing Splash Lipidomix (Avanti) as internal standards.

#### 2.2.4. Metabolomics LC/MS Methods

Metabolomics samples were analyzed using two separate LC-MS methods on Vanquish UPLCs coupled to Q-Exactive Plus mass spectrometers. Metabolites analyzed in positive ionization mode were separated using a SeQuant^®^ ZIC^®^-pHILIC column (5 μm, 200 Å, 150 × 2.1 mm). Mobile phase A was 20 mM ammonium carbonate in water (pH 9.2) and mobile phase B was acetonitrile. The flow rate was 150 μL/min and the gradient was t = −6, 80% B; t = 0, 80% B; t = 2.5, 73% B; t = 5, 65% B, t = 7.5, 57% B; t = 10, 50% B; t = 15, 35% B; t = 20; 20% B; t = 22, 15% B; t = 22.5, 80% B; t = 24; 80% B. The mass spectrometer was operated in positive ion mode using data-dependent acquisition (DDA) mode with the following parameters: resolution = 70,000, AGC target = 3.00 × 10^6^, maximum IT (ms) = 100, scan range = 70–1050. The MS2 parameters were as follows: resolution = 17,500, AGC target = 1.00 × 10^5^, maximum IT (ms) = 50, loop count = 6, isolation window (*m*/*z*) = 1, (N)CE = 20, 40, 80; underfill ratio = 1.00%, Apex trigger(s) = 3–10, dynamic exclusion(s)  =  25.

Metabolites analyzed in negative ionization mode were separated using a reverse phase ion-pairing chromatographic method using an Agilent Extend C18 RRHD column (1.8 μm, 80 Å, 2.1 × 150 mm). Mobile phase A was 10 mM tributylamine, 15 mM acetic acid in 97:3 water:methanol pH 4.95; mobile phase B was methanol. The flow rate was 200 μL/min and the gradient was t = −4, 0% B; t = 0, 0% B; t = 5; 20% B; t = 7.5, 20% B; t = 13, 55% B; t = 15, 95% B; t = 18.5, 95% B; t = 19, 0% B; t = 22, 0% B. The mass spectrometer was operated in negative ion mode using data-dependent acquisition (DDA) mode with the following parameters: resolution = 70,000, AGC target = 1.00 × 10^6^, maximum IT (ms) = 100, scan range = 70–1050. The MS/MS parameters were as follows: resolution = 17,500, AGC target = 1.00 × 10^5^, maximum IT (ms) = 50, loop count = 6, isolation window (*m*/*z*) = 1, (N)CE = 20, 50, 100; underfill ratio = 1.00%, Apex trigger(s) = 3–12, dynamic exclusion(s) = 20. RAW files were converted to mzML files using msconvert from ProteoWizard, using vendor centroiding on all scans.

We have deposited our mzML files into the Metabolights database [21], with accession number MTBLS3097.

#### 2.2.5. Lipidomics LC/MS Methods

Lipidomics samples were analyzed using two separate injections, one for positive mode ionization and one for negative mode ionization, both using a Vanquish UPLC coupled to a Q-Exactive Plus mass spectrometer. Lipids were separated using a Thermo Scientific Accucore C30 column (2.6 μm, 150 Å, 2.1 × 250 mm). Buffer A was 20 mM ammonium formate in 60:40 acetonitrile:water + 0.25 μM medronic acid and buffer B was 20 mM ammonium formate in 90:10 isopropanol:acetonitrile + 0.25 μM medronic acid. 

For both ionization modes, the chromatography was as follows: the flow rate was 0.2 mL/min, and the gradient was t = −7, 30% B, t = 0, 30% B, t = 7, 43% B, t = 12, 65% B, t = 30, 70% B, t = 31, 88% B, t = 51, 95% B, t = 53, 100% B, t = 55, 100% B, t = 55.1, 30% B, t = 60, 30% B.

For both positive and negative ionization modes, the mass spectrometry settings were as follows: data-dependent acquisition (DDA) was performed with the following parameters: resolution = 140,000, AGC target = 3.00 × 10^6^, maximum IT (ms) = 100, scan range = 200–2000. The MS2 parameters were as follows: resolution = 17,500, AGC target = 3.00 × 10^6^, maximum IT (ms) = 150, loop count = 8, isolation window (*m*/*z*) = 1, (N)CE = 20, 30, 40; underfill ratio = 1.00%, Apex trigger(s) = 5–30, dynamic exclusion(s) = 15 s. RAW files were converted to mzML files using msconvert from ProteoWizard, using vendor centroiding on all scans. We have deposited our mzML files into the Metabolights database, with accession number MTBLS3097.

#### 2.2.6. External Lipidomics Datasets

Pooled samples of NIST SRM1950 plasma analyzed on an Agilent 6546 QTOF instrument from [22] were retrieved by following the instructions provided in the manuscript. RAW files were converted to mzML files using ProteoWizard, using vendor centroiding on all scans. Lipid spectral libraries described in [22], were retrieved in .msp format following the instructions provided in the manuscript.

## 3. Results and Discussion

### 3.1. Updated GUI Facilitates Investigation of Large LC-MS/MS Datasets

Since the last publication of MAVEN [23], many UI, infrastructural, and algorithmic changes have been made (Figure 2).

Users can load and search multiple fragmentation libraries simultaneously.Users can browse through individual compounds—automatically extracting EICs, displaying location of MS/MS events, and browse through all fragmentation spectra, which are sortable by retention time and spectral purity throughout the loaded dataset. Peaks in the raw data may be manually selected to trigger a targeted search of the peak of interest (in real time a consensus MS/MS spectrum is calculated, spectral libraries are scanned, and MS/MS similarity scores are computed). Spectral similarities are displayed in a dedicated MS/MS widget. Peaks may be quantified a number of different ways, or manually integrated. Manually integrated peak groups can be saved to a special “Bookmarks” table.Peak groups may be tagged with user-configurable labels. Peak groups may take on an unlimited number of labels, and peak group results tables may be filtered based on label contents for efficient navigation and visualization.Internal infrastructure has been improved to save projects, sample information, and search results into SQLite databases. This makes it possible to interact with project files outside of MAVEN2, through both SQLite database visualization programs and programmatically via many mainstream programming languages.

A recurring theme in MAVEN2’s user experience is the proximity of calculated results to the raw data associated with results. For instance, spectra and EIC visualization is featured prominently, encouraging manual verification. Any identification or annotation may be validated by hand, and the raw data associated with such an identification may be explored deeply. All spectral data associated with an experiment is loaded in MAVEN2 at the time an experiment is opened, which makes raw data very fast to access. Processing of experimental data, e.g., performing spectral library searches, is also engineered to be efficient, encouraging on-the-fly reprocessing. Above all, the tool is meant to be usable, accessible, and interactive. For a complete description and link to a video tutorial, see example analysis script and corresponding report (Appendix A).

### 3.2. Comparison of Fragmentation Matching Scoring Methods

To allow for various user case scenarios, we have implemented a number of MS/MS matching algorithms. In cases where library spectra contain intensity information, dot-product and fraction of TIC matched can be used for scoring. In cases where only the *m*/*z* (and not the intensity) of fragments are known, (for example, purely in silico fragmentation libraries), other approaches are available, such as number or fraction of fragments matched, as well as the hypergeometric or MVH scoring approaches.

To evaluate the efficacy of the various MS/MS scoring methods implemented in MAVEN2, we constructed a gold standard dataset using spiked-in compounds. Standard compounds were spiked into samples containing either a water or yeast background. We noted the correct retention time of standards, which allowed us to assign peak groups to true positive matches in the yeast samples (see Appendix A). We then compared observed MS/MS spectra of annotated peak groups to standard compound spectra contained in a previously generated spectral library, using a variety of scoring methods. We explored precision and recall across these score types (Appendix A). The hypergeometric score achieved the best precision across a large range of score thresholds, especially in the yeast background samples (Figure 3). For all scoring methods, we found that as the score threshold increased, the total number of identified compounds decreased, though the exact counts of correct identifications and incorrect identifications varied between scoring approaches (see Appendix A).

We found that sample background had a dramatic effect on performance, regardless of the scoring approach. Due to the increased complexity of the yeast background, there were more peak groups detected, which provided more opportunities for false matches. Additionally, some standard compounds did not fragment well in our experimental workflow, limiting the amount of information in their library spectra, resulting in lower match scores. We suggest that the yeast background data are more representative of what a typical user might encounter in a biological sample, emphasizing the practical utility of the hypergeometric score as a spectral matching approach.

Overall, we were encouraged by our results, which suggest that the MAVEN2 MS/MS-based workflow may be useful in discriminating authentic from spurious compound identifications. It is instructive to review some of the cases that could not be properly categorized. Specifically, when fragmentation spectra are very similar between isomers, our algorithm struggles to consistently select the correct metabolite among isomers. This is especially a problem when no observed MS/MS spectrum matches especially well with the library spectrum (Appendix A). Our finding supports the previous observation that fragmentation spectra alone may not be sufficient for correct identification [12]. Knowledge of retention time may be helpful in cases where isoforms can be chromatographically separated, however, this requires establishing elution times through purified standards, which may not be feasible for large sets of metabolites. The code used to carry out this analysis is available in Appendix A.

### 3.3. CalicoLipids Libraries Return Unique Lipid Identifications

We performed a series of lipidomics searches using lipidomics datasets both generated in-house and retrieved from publicly available repositories. All datasets were searched against both our novel CalicoLipids in silico fragmentation library and the publicly available MS-DIAL library. Dataset features were annotated as specific lipid species using the hypergeometric score with library-specific score thresholds (see Methods and Appendix A), and classified according to the lipid annotations assigned by the CalicoLipids and MS-DIAL libraries.

Applying this approach to all dataset features, we found 12,054 unique features detected from all samples (Table 1), with approximately 21% (2612) could be annotated by at least one library. A significant number of features were annotated by only a single library, (1009 by CalicoLipids, 461 by MS-DIAL), and about 50% of all annotated features were annotated by both libraries (1142). Among features that were annotated by both libraries, we observed strong annotation agreement: approximately 94% percent of annotations (1078/1142) were in agreement on lipid class level, and approximately 87% (990/1142) agreed on lipid class, summed composition, and adduct form. Stratification of feature and compound identifications by dataset revealed that CalicoLipids returned more identifications than MS-DIAL independent of instrument or ionization mode (Table 2). 

Given the differences in identifications rendered by each library, we were curious about the representation of different lipid classes by the CalicoLipids and MS-DIAL libraries. To this end, we enumerated for each of the CalicoLipids and MS-DIAL libraries the total number of library entries and annotated features associated with each lipid class and adduct form combination (Appendix A). An examination of the ten most abundant lipid classes found in our dataset revealed some interesting trends (Figure 4). For instance, CalicoLipids identified many more sphingomyelins (SMs) than MS-DIAL; however, the CalicoLipids library contains many more spectra for this class than MS-DIAL (14,076 vs. 117, see Appendix A). Both libraries appear to have their areas of strength; however, for instance, MS-DIAL could identify many more Ceramides in the Agilent samples than CalicoLipids (where the CalicoLipids library contains 36,288 Ceramide spectra and the MS-DIAL library contains 39,882 spectra, see Appendix A). It is worth mentioning that some lipid classes were represented only in one library, such as for instance, the oxidized phosphatidylcholines (OxPCs) were present only in MS-DIAL, while the cholesterol esters (CEs) were present only in CalicoLipids (for a complete list, see Appendix A). A complete script of all lipidomics analyses, with additional detailed information, is available in Appendix A.

## 4. Conclusions

Mass spectrometry is a fast-evolving field, and as the technology changes, so too must the software used to analyze mass spectrometry data. Here we describe new developments in the mass spectrometry analysis program MAVEN2 (an updated version of MAVEN) to access, analyze, and investigate LC-MS and LC-MS/MS data. The program has been redesigned and includes new features such as spectral library searching, MS/MS-based scoring, and user-friendly visualizations.

The ability to identify compounds based on MS/MS spectra makes MAVEN2 a valuable tool for analyzing lipidomics datasets. To that end, we created our own in-silico lipid spectral libraries, which we used in MAVEN2 to search several lipidomics datasets. We also carried out similar lipidomics searches using publicly available lipid libraries. Evaluating performance is challenging without gold standards, so we developed an approach based on determining library-specific score thresholds, which allowed for a fair comparison of our library versus others. We showed that many lipids were identified by only one lipid library, suggesting that our library may be complementary to existing in-silico lipid spectral libraries. However, we also discovered that the bulk of features remained unannotated by either library, suggesting an opportunity for future work.

All libraries, codebases, and downloadable MAVEN2 executable applications are released under open-source licenses and are freely available for download at https://github.com/eugenemel/maven/releases/latest (accessed 23 July 2022).

## Figures and Tables

**Figure 1 metabolites-12-00684-f001:**
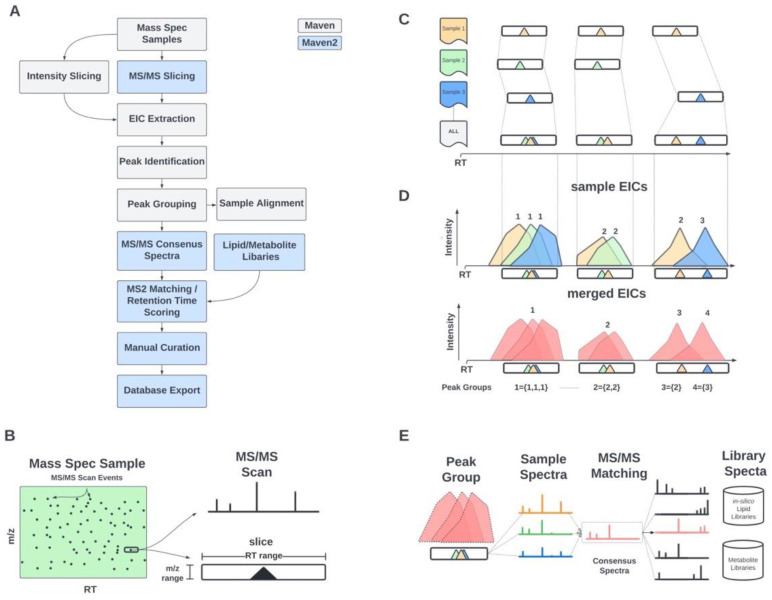
Overview of MS/MS-based workflow in MAVEN2. MAVEN2 implements MS/MS-based slicing, construction of consensus spectra, and spectral library matching. Outline of key steps and novel algorithmic implementations are highlighted in blue (**A**). MS/MS scans collected in LC/MS run are used as seeds for formation of “slices”—blocks in *m*/*z* and retention time (RT) space surrounding the MS/MS scan’s precursor *m*/*z* (**B**). Slices from all samples are merged based on overlaps in *m*/*z* or RT space (**C**). These merged slices are used to generate extracted ion chromatograms (EICs) and summed to form a merged EIC. Peak groups are defined by overlapping regions of intensity in merged EIC. The peaks picked from individual sample EICs are then associated with their corresponding peak groups (**D**). MS/MS Spectra corresponding to a peak group are combined to form a consensus spectrum, which is then searched against spectral libraries to identify compounds (**E**). This approach allows for annotation of peaks in samples even when no MS/MS spectra were collected.

**Figure 2 metabolites-12-00684-f002:**
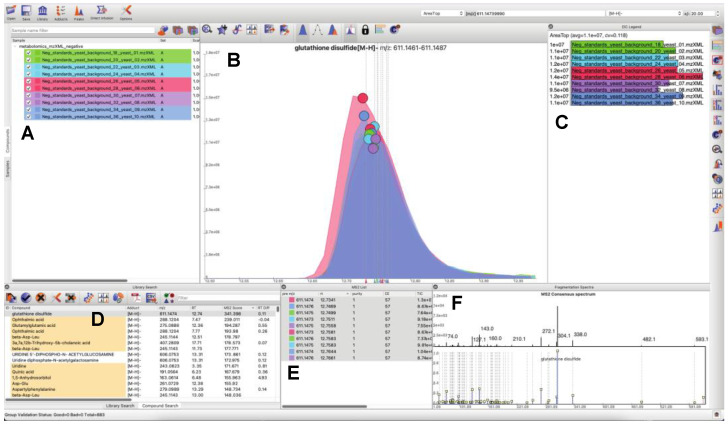
MAVEN2 User Interface. The MAVEN2 user interface presents all relevant information for compound validation in a single view. This view includes a list of loaded samples (**A**), overlayed EICs from multiple samples (**B**), sample-specific quantitation information (**C**), the result of a fragmentation spectral library search (**D**), a list of MS/MS events associated with a putative identification (**E**), and spectral matches between a consensus MS/MS spectrum and the library spectrum (**F**). Shown above is a match to a library spectrum of glutathione disulfide.

**Figure 3 metabolites-12-00684-f003:**
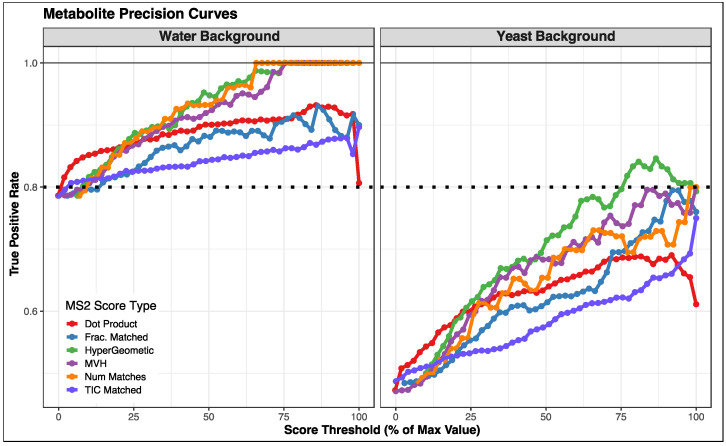
Metabolomics Spike-In Standards Precision Curve. True positive rate as a function of score threshold was assessed in either water or yeast backgrounds. At each threshold of MS/MS score we calculated the fraction of matches that were correctly matched (based on known retention time of spiked-in standards, see Appendix A). To allow for direct comparison between different scoring methods, the X-axis is scaled to the maximum value of each scoring method. As expected, precision was worse in the yeast background compared to the water background due to the presence of a higher number of background peaks. Performance of all methods improved with higher thresholds, with the hypergeometric score having the best performance (in terms of precision).

**Figure 4 metabolites-12-00684-f004:**
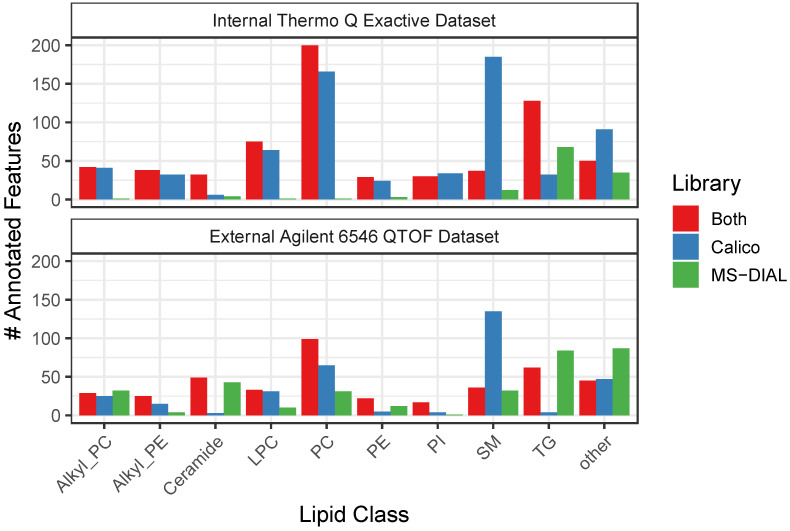
Summary of identified lipids by lipid class, dataset, and library. Counts of annotated features organized by lipid class, instrument dataset, and in-silico library. “Both” refers to features annotated as the same class by both CalicoLipids and MS-DIAL libraries. “Calico” and “MS-DIAL” refer respectively to features annotated exclusively by one library. The ten most commonly annotated classes are shown, along with all other classes represented in the “other” category. For example, the CalicoLipids library identifies many more SMs in both the internal Thermo and external Agilent datasets (SM, blue, upper and lower plots), while MS-DIAL identifies more TGs in the external Agilent dataset (TG, green lower plot).

**Table 1 metabolites-12-00684-t001:** Lipid annotation agreement across several lipid datasets. All features from all lipid datasets were searched against both the CalicoLipids and MS-DIAL lipid libraries. Among the 12,054 features found in all datasets, 2612 (21.7%) could be annotated by a lipid library. Features that could be annotated by both lipid libraries demonstrated varying degrees of agreement in their annotations. As the level of annotation agreement increased, the number of features with the corresponding level of agreement decreased. In a small number of cases (44 of 12,054), each library suggested very different annotations (“full disagreements”). For cases where there was a single, unambiguous annotation (either by single-library annotation or complete agreement of multiple library annotations), the number of identified compounds could also be determined. The high number of features as compared to compounds among the full agreements indicate that for many compounds, multiple adduct forms were detected by both libraries.

Annotation Type	# Features (%)	# Compounds
All Features	12,054 (100%)	
Unannotated	9442 (78.30%)	
Identified only in CalicoLipids library	1009 (8.40%)	768
Identified only in MS-DIAL library	461 (3.80%)	429
Identified by both libraries	1142 (9.50%)	
Identified by both libraries with same lipid class	1078 (8.90%)	
Identified by both libraries with same lipid class and adduct	994 (8.20%)	
Identified by both libraries with same summed composition and adduct	990 (8.20%)	
Identified by both libraries as same compound (full agreement)	512 (4.20%)	253
Identified by both libraries with different lipid class, adduct, summed composition, and compound (full disagreement)	44 (0.4%)	

**Table 2 metabolites-12-00684-t002:** Lipid identifications discovered in individual datasets. Two datasets, an internal and external dataset (“Internal Thermo Q Exactive Dataset” and “External Agilent 6546 QTOF Dataset”) were analyzed in both positive and negative ionization mode, and separately searched with MAVEN2 using either the CalicoLipids or MS-DIAL spectral libraries. We found that a higher proportion of the features could be identified in the internal dataset than in the external dataset, irrespective of spectral library or ionization mode. We were able to identify more compounds using the CalicoLipids library vs the MS-DIAL library for every search, with the exception of the negative mode samples from the External Agilent 6546 QTOF Dataset (245 compounds vs. 362 compounds, respectively).

Dataset	Library	Ionization Mode	# Features	# Features Identified (%)	# Compounds Identified
Internal Thermo Q Exactive Dataset	CalicoLipids	positive	3412	995 (29.20%)	741
	MS-DIAL	positive	3412	525 (15.40%)	442
	CalicoLipids	negative	1301	384 (29.5%)	300
	MS-DIAL	negative	1301	304 (23.40%)	263
External Agilent 6546 QTOF Dataset	CalicoLipids	positive	4162	502 (12.10%)	415
	MS-DIAL	positive	4162	378 (9.10%)	358
	CalicoLipids	negative	3179	270 (8.50%)	245
	MS-DIAL	negative	3179	396 (12.50%)	362

## Data Availability

Raw files associated with this study have been deposited in the Metabolights database with accession number MTBLS3097.

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
