# Peer review of "MAVEN2: An Updated Open-Source Mass Spectrometry Exploration Platform"

_metabolites, 2022, doi:10.3390/metabo12080684_

Round 1

Reviewer 1 Report

MAVEN is a transformative software package that enables stable-isotope labelling and untargeted metabolomics. Here the authors make significant additions to the software (MAVEN2), namely enabling MS/MS spectral library matching using a suite of scoring approaches and the additional capacity/workflow to generate in silico spectral libraries (using lipidomics as the test case).

The contribution is significant and will greatly benefit the field. I only have some minor queries and suggestions.

1.      Figure 1 is mislabelled ‘MAVEN 9’ and misspelled ‘curration’

2.      RPB – the CalicoLipid + MSDial agreement/disagreement distributions are quite different. It appears there are more agreements at low hypogeometric scores for the Calico. Do you think this is a function of using a hypogeometrtic score at the MS2 for compounds (lipids) which often have few MS2 fragments (say compared to peptide MS2 spectra)?

3.      Following on from the previous. When using the METLIN MS2 database, it is often the case that metabolites/lipids only have 2-3 MS2 fragment ions. When calculating the hypergeometric score, how is the ‘reference spectrum’ selected? And does it contain a similar number of fragment ions?

4.      The generation of the ‘super pool’ of polar metabolites was unclear. Could the authors please clarify the approach/intention.

5.      I strongly recommend the authors consider making a short (10-15min) video detailing how to use the new features of MAVEN2. This shouldn’t be necessary for publication but I think it would greatly enhance the accessibility of the updated package and new features. For example, this user would be very interested in creating their own spectral library (from in-house standards) for routine metabolite ID. It may well be straightforward but having additional resources to avoid any pitfalls (e.g. using a dot product score and utilise the fragment ion intensity – instead of the HGS- for authentic standard spectral libraries).

Author Response

1.      Figure 1 is mislabelled ‘MAVEN 9’ and misspelled ‘curration’
> We have this correction in the updated submission.

2.      RPB – the CalicoLipid + MSDial agreement/disagreement distributions are quite different. It appears there are more agreements at low hypogeometric scores for the Calico. Do you think this is a function of using a hypogeometrtic score at the MS2 for compounds (lipids) which often have few MS2 fragments (say compared to peptide MS2 spectra)?
> The reason there are more agreements at low hypergeometric scores for the CalicoLipids library as compared to the MS-DIAL library likely has to do with fundamental differences between the fragments contained in the CalicoLipids and MS-DIAL libraries.  The existence of agreements that have low hypergeometric scores in one library and high hypergeometric scores in another suggests that key fragments may be missing from the library that produced the lower score (assuming that the agreement corresponds to a correct identification).  This suggests that reviewing the spectral content of different libraries, particularly when comparing agreements, may help in synthesizing better spectral libraries for the future.  This phenomenon also underscores the value of searching datasets against multiple spectral libraries.
> The reviewer's comment inspired us to add an analysis to the manuscript, where we have enumerated the number of fragments per compound ion in each of the CalicoLipids and MS-DIAL spectral libraries. We have added this analysis to Supplementary 1 and Supplementary 2.  found that on average, CalicoLipids has 11 fragments per compound ion, while MS-DIAL has 6.7 fragments/compound ion. The existence of more fragments per compound ion in CalicoLipids is consistent with our finding that the threshold generated by the RBP method should be higher for CalicoLipids than MS-DIAL: hypergeometric scores reward absolute numbers of fragments, so the detection of more fragments produces a higher score.

3.      Following on from the previous. When using the METLIN MS2 database, it is often the case that metabolites/lipids only have 2-3 MS2 fragment ions. When calculating the hypergeometric score, how is the ‘reference spectrum’ selected? And does it contain a similar number of fragment ions?
> We understand 'reference spectrum' to refer to the library spectrum for a given compound ion.  In the manuscript, we describe how we generate library spectra for the CalicoLipids library in the methods section "Generation of in silico lipid spectral library".  We also briefly describe how the metabolite spectral library is created in the methods section "Construction of polar metabolite spectral library".
> In both cases, the generation of MS/MS spectra from chemical standards informed the creation of reference spectra. For the metabolomics spectral library, chemical standards were loop-injected, and measured at various collision energies.  Multiple observed scans were combined into consensus spectra.  Fragments that were sufficiently intense and reproducible across scans were retained to form library spectra.
> The in silico lipidomics library leveraged standards as well as previously described fragmentation information available in the literature.  We sought to include fragments that could be observed with an intensity as low as 0.2% of the intensity of the most abundant peak. Previously, others have carried out a similar exercise, using an intensity threshold of 1.0% of the intensity of the most abundant peak.  We found that using our lower intensity threshold of 0.2% resulted in the inclusion of many more fragment peaks, and we believe, much more useful information.
> The reviewer's comment about there being only 2-3 MS2 fragment ions for certain entries is a good point - in some cases, there just isn't much information in an MS2 spectrum.  This exists in contrast to bottom-up proteomics, where peptides typically produce rich fragmentation spectra (for instance, tryptically digested peptides may have lengths of approximately 6-30 amino acids, where many b- and y- ions might be observed corresponding to breaking any of the amide bonds present in the peptide's backbone).
> We believe the hypergeometric score is an especially valuable scoring approach because it directly addresses the reviewer's concern.  In cases where there are not many spectral peaks to be found, the hypergeometric score will not be very high, reflecting the fundamental dearth of spectral information.  However, in cases where there are many spectral peaks to be found, the hypergeometric score will be much higher.  Consider two cases, metabolites X and Y.  Metabolite X contains only 3 fragments in its library spectrum, while Metabolite Y contains 9 fragments in its library spectrum.  Imagine that spectral matches are found for both Metabolites X and Y in some observed data, matching 2/3 of the reference peaks.  For Metabolite X, this means that 2 of 3 peaks are matched, while for metabolite Y, 6 of 9 peaks are matched.  While both of these have identical fractions of their reference peaks matched (2/3), Metabolite X has a hypergeometric score of 18.5529 while Metabolite Y has a hypergeometric score of 49.7997. According to the hypergeometric score, we can feel much more confident in the authenticity of the Metabolite Y identification, as compared to the Metabolite X identification.

4.      The generation of the ‘super pool’ of polar metabolites was unclear. Could the authors please clarify the approach/intention.
> We have restructured the "Generation of metabolomics datasets" section to include more clarification and explanation surrounding the super pool approach.

5.      I strongly recommend the authors consider making a short (10-15min) video detailing how to use the new features of MAVEN2. This shouldn’t be necessary for publication but I think it would greatly enhance the accessibility of the updated package and new features. For example, this user would be very interested in creating their own spectral library (from in-house standards) for routine metabolite ID. It may well be straightforward but having additional resources to avoid any pitfalls (e.g. using a dot product score and utilise the fragment ion intensity – instead of the HGS- for authentic standard spectral libraries).
> We appreciate the suggestion, and have created a tutorial video, which we have uploaded to youtube.  We have added the youtube link to Supplementary 3, and now mention that a tutorial video is available in the main text.
The tutorial video is available at this URL:
https://youtu.be/QUSX0GJ6Gsk

Reviewer 2 Report

This manuscript presents an important update to the open-source MAVEN software (now called MAVEN2), which includes new features such as spectral library searching, MS/MS scoring, user-friendly visualizations - and the ability for users to import pre-existing NIST msp libraries. The authors also demonstrated the utility of the MAVEN2 software through the generation of lipidomics datasets. Additionally, the use of well-established scoring algorithms, including hypergeometric score (HGS) and multiple class weighted hypergeometric score (MVH) was of great significance. I recommend that the manuscript be accepted as is, with no required changes needed.

Author Response

Thank you for taking the time to review the manuscript, and for your kind words.